# The Ubiquitin–Proteasome System in Immune Cells

**DOI:** 10.3390/biom11010060

**Published:** 2021-01-05

**Authors:** Gonca Çetin, Sandro Klafack, Maja Studencka-Turski, Elke Krüger, Frédéric Ebstein

**Affiliations:** Institute of Medical Biochemistry and Molecular Biology, Universitätsmedizin Greifswald, D-17475 Greifswald, Germany; Gonca.Cetin@med.uni-greifswald.de (G.Ç.); Sandro.Klafack@med.uni-greifswald.de (S.K.); Maja.Studencka-Turski@med.uni-greifswald.de (M.S.-T.); ebsteinf@uni-greifswald.de (F.E.)

**Keywords:** ubiquitin–proteasome system, proteostasis, immunity, autoinflammation, cancer

## Abstract

The ubiquitin–proteasome system (UPS) is the major intracellular and non-lysosomal protein degradation system. Thanks to its unique capacity of eliminating old, damaged, misfolded, and/or regulatory proteins in a highly specific manner, the UPS is virtually involved in almost all aspects of eukaryotic life. The critical importance of the UPS is particularly visible in immune cells which undergo a rapid and profound functional remodelling upon pathogen recognition. Innate and/or adaptive immune activation is indeed characterized by a number of substantial changes impacting various cellular processes including protein homeostasis, signal transduction, cell proliferation, and antigen processing which are all tightly regulated by the UPS. In this review, we summarize and discuss recent progress in our understanding of the molecular mechanisms by which the UPS contributes to the generation of an adequate immune response. In this regard, we also discuss the consequences of UPS dysfunction and its role in the pathogenesis of recently described immune disorders including cancer and auto-inflammatory diseases.

## 1. The Ubiquitin–Proteasome System (UPS): Structure and Function

Four decades ago, it was discovered that conjugation of intracellular proteins with the small highly conserved ubiquitin polypeptide leads to their subsequent degradation by 26S proteasomes [1,2,3,4]. Since then, our understanding of the function of the ubiquitin–proteasome system (UPS) has progressed continuously. It is now well established that ubiquitination is mediated by an enzyme cascade (i.e., E1, E2, and E3) which transfers ubiquitin molecules to cellular targets in a sequential manner [5,6]. As illustrated in Figure 1, a ubiquitin-activating enzyme (E1) first activates ubiquitin in an ATP-hydrolyzing reaction which is then transferred to a cysteine residue of the active site of a ubiquitin-conjugating enzyme (E2). Then, in a two-step reaction, E2 enzymes promote ubiquitin conjugation to substrates with the help of ubiquitin-ligases (E3) which recognize specific targets. In this process, substrate specificity is ensured by an estimated number of 500–1000 putative E3 ligases encoded by the human genome [7]. Canonical ubiquitination is defined by the formation of isopeptide bonds between lysine residues of target proteins and the C-terminal glycine (glycine 76) of ubiquitin [8]. Non-canonical ubiquitination is less frequent and may occur on serine/threonine or cysteine residues of substrates via ester or thioester bonds, respectively [9,10,11,12,13]. Proteins may be marked with one single ubiquitin moiety on one single site resulting in mono-ubiquitination which is widely recognized as a major molecular signal regulating chromatin remodeling as well as protein sorting and/or trafficking [14,15]. However, the attached ubiquitin molecule itself exhibits eight potential ubiquitination sites (M1, K6, K11, K27, K29, K33, K48 and K63) which may be used for further ubiquitination cycles, thereby giving rise to polyubiquitin chains.

Depending on the ubiquitin residue used for poly-ubiquitination, the assembled ubiquitin chains present various topologies which behave differently and carry distinct biological fates for the modified substrate. It is well appreciated that ubiquitin moieties linked to each other via lysine at position 48 (K48) represent the canonical signal for proteasomal degradation [16], while those linked via K63 allow the clearance of damaged organelles (i.e., ribosomes, endoplasmic reticulum, and/or mitochondria) by the autophagy-lysosome pathway [17]. Other ubiquitin linkages are primarily viewed as non-proteolytic signals involved in the regulation of various cellular processes with DNA repair and cell signaling being the most prominent ones [18]. Proteins modified with K48-linked polyubiquitin chains are typically degraded by the 26S proteasome which is a 2.5 MDa complex consisting of two sub-complexes, namely, the 20S core particle (CP) and the 19S regulatory particle (RP) [19,20]. The 19S RP plays a crucial role in the recognition of ubiquitin-modified proteins as well as their deubiquitination and subsequent unfolding before degradation by the 20S CP [21,22]. As shown in Figure 1, the 20S CP exhibits a barrel-like structure made out of four heptameric rings which are themselves composed of α- and β-subunits. The two outer rings comprise seven α-subunits whose N-termini are oriented towards seven β-subunits forming the two inner rings. The catalytic activity of the 20S complex is ensured by the incorporation of the β1, β2, and β5 subunits which are threonine proteases exhibiting caspase-, trypsin-, and chymotrypsin-like activities, respectively [23,24]. Under certain conditions, alternative catalytic subunits may assemble into newly synthetized proteasomes to build so-called immunoproteasomes (IP) that carry β1i, β2i, and β5i inducible subunits instead of the β1, β2, and β5 standard subunits [25,26]. IP distinguish themselves from their standard counterparts by exhibiting higher proteolytic activities which make them, under stress conditions, more efficient at degrading ubiquitin-modified proteins [27,28,29,30,31]. It is widely accepted that IP are constitutively expressed in immune cells, while they are induced in non-immune tissues in response to type I and/or II interferons (IFN) [26]. Mixed-type (also referred to as intermediate-type) proteasomes are defined as partially assembled IP bearing one or two out of the three inducible subunits and may be found under normal conditions in some tissues with high protein turnover rates such as the liver [32,33,34,35]. Complexity to this system is further added by the observation that 20S CP may be reversely capped at either one or both ends with proteasome regulators such as an 11S complex composed of PA28αβ or PA28γ subunits (Figure 1), PA200 or PI31 [36,37,38,39,40]. Because of these many possible combinations, an enormous diversity of proteasomes arises, whose biological relevance is, however, not fully understood. The majority of proteasomes resides in both the cytosol and nucleus, while damaged proteasomes are targeted to lysosomes through macro-autophagy [41] or removed extracellularly by means of exosomes [42]. Intriguingly, it was recently suggested that proteasomes are present in the lumen of endocytic and phagocytic organelles of professional antigen-presenting cells [43].

## 2. The UPS Is a Major Regulator of Cell Signalling in Response to Pathogens

Thanks to its ability to specifically eliminate intracellular proteins in response to diverse stimuli, the UPS is involved in almost all aspects of cell physiology and development [44]. The broad range of functions of the UPS is particularly well exemplified in immune cells of myeloid origin whereby it regulates key features of both innate and adaptive immunity including signal transduction, protein homeostasis, and MHC class I antigen presentation. Myeloid cells are critical components of the immune system and include monocytes, macrophages, (DC), and granulocytes whose main function resides in patrolling the body for potential invaders [45,46]. Recognition of microbial products by myeloid cells occurs through the binding of pathogen-associated molecular patterns (PAMP) to their pattern recognition receptor (PRR). Depending on their cellular localization and ligand specificity, PRR are traditionally subdivided into Toll-like receptors (TLR), retinoic acid-induced gene-I (RIG-I)-like receptors (RLR), nucleotide-binding oligomerization domain (NOD)-like receptors, and cytosolic DNA or RNA sensors inducing the cyclic-GMP-AMP synthase (cGAS) and protein kinase R (PKR), respectively [46]. The sensing of PAMP by PRR initiates a series of signalling cascades which ultimately leads to the activation of major transcription factors driving inflammation, namely, the nuclear factor-κB (NF-κB) as well as the interferon regulatory factors (IRF) 3 and 7 [47,48]. Following their translocation into the nucleus, NF-κB and IRF3/IRF7 induce the transcription of proinflammatory cytokines (i.e., TNF-α, IL-1β, and IL-6) and interferons (IFN), respectively, to counteract the invading pathogens. Strikingly, a large body of work, notably by the groups of Karin and Dixit, has shown that the intracellular signalling pathways triggered by PRR in response to danger signals heavily relies on the UPS which tightly controls key intersections of the initiated cascades [49,50,51,52].

In this process, the breakdown of signalling components by the UPS is intricately associated with the generation of non-proteolytic ubiquitin chains, among which are K63-linkages that regulate protein activities and/or interactions. For instance, as shown in Figure 2, during infection with gram-negative bacteria, binding of LPS to TLR-4 on myeloid cells results in the activation of the E3 ubiquitin ligase TRAF6 which generates K63-linked polyubiquitin chains, thereby activating the downstream TAK1 complex [53,54]. One major role of TAK1 is to activate the canonical heterotrimeric IκB kinase (IKK) complex which itself consists of the related protein kinases IKKα and IKKβ as well as IKKγ, a regulatory component also referred to as NF-κB essential modulator (NEMO) [55,56]. Importantly, TRAF6 is also implicated in the activation of the so-called linear ubiquitin chain assembly complex (LUBAC) comprising the proteins HOIP, HOIL-1, and sharpin [57,58]. Linear ubiquitination of NEMO by LUBAC promotes a NEMO conformational change that facilitates IKK assembly [59,60]. Full activation of IKK requires phosphorylation of IKKα (ser 176 and 180) and IKKβ (ser 177 and 181) by TAK1 [61]. One major substrate for IKK (i.e., IKKβ) is IκBα, the inhibitory component of NF-κB. Indeed, IκB*α* phosphorylation at Ser32 and Ser36 by IKK unmasks a lysine residue that is used by the E3 ubiquitin ligase SCF*^TRCP^*, thereby catalysing the K48-linked ubiquitination of IκBα and promoting its proteasomal degradation [62,63]. Interestingly, it has been suggested that IκBα is faster degraded by IP [27,64,65] than standard proteasomes (SP). Likewise, hybrid (immuno)-proteasomes (HP) asymmetrically capped at each end with 19S and PA28 complexes (Figure 1) have been shown to increase IκBα protein turnover [66,67]. Given that both IP and PA28 are constitutively present in myeloid cells [26,68], these observations propose a seductive rationale for the rapidity of these cells to release proinflammatory cytokines in response to pathogens. This assumption is in line with a number of reports showing that blocking the β5i inducible subunit with the small molecule inhibitor ONX 0914 (formerly PR-957) results in decreased production of proinflammatory cytokines in various disease models [69,70,71,72,73,74,75]. Similar observations were made in knockout mice lacking β5i in other various inflammation models [65,70,76,77]. Whether the beneficial effects detected on inflammation following IP inhibition really rely on impaired NF-κB signalling remains, however, a matter of debate. Some studies have highlighted the decreased efficiency of proteasomes devoid of functional β5i to degrade IκB*α* [27,64,65], while other investigations not [70,78]. The reasons for these discrepancies are unclear but may reflect distinct experimental conditions. Importantly, besides inducing proinflammatory genes, NF-κB also promotes the upregulation of the ubiquitin hydrolases TNFAIP3 (TNFα-induced protein 3, also known as A20), CYLD, and OTULIN which hydrolyse K63- and Met1-ubiquitin chains with varying efficiencies (Figure 2) [79,80,81,82,83]. By removing ubiquitin chains from TRAF6 and IKK, these deubiquitinating enzymes ensure a negative feedback loop in NF-κB signalling.

Apart from the NF-κB transduction pathway, ubiquitination is also a key regulatory mechanism for IFN production in response to viruses. As illustrated in Figure 3, one central component in this signalling cascade is the NEMO-TBK1/IKKε complex which catalyses the phosphorylation of the transcription factors IRF3 and/or IRF7 for nuclear translocation. The assembly of NEMO-TBK1/IKKε requires the generation of multiple non-proteolytic ubiquitin chains. These include notably K63-linked polyubiquitin chains catalysed by the E3 ubiquitin ligase TRAF3 following recognition of viral products by some intracellular TLR, such as TLR3. TLR7, TLR8 or TLR9, and/or RIG-I-like receptors [84,85,86].

Suppression of TLR/RLR signalling in response to viral products involves the generation of K48-linked polyubiquitin chains which target key components of the cascade for proteasomal degradation. Such modifications affect notably IRF3/IRF7 and TBK1 by the E3 ubiquitin ligases RAUL [87,88] as well as DTX4 [89], TRIM27 [90] or TRIP [91]. In addition, target genes of IRF3/IRF7 include ubiquitin hydrolases such as *CYLD*, *USP21*, *OTUD5,* and *USP3* which trim non-proteolytic chains [92,93,94], thereby deactivating key signalling nodes and preventing sustained type I IFN production (Figure 3). Another negative regulatory mechanism in the absence of danger signals is the constitutive K48-ubiquitination of cGAS which results in its degradation by autophagy [95]. Upon DNA virus infection, cGAS becomes rapidly stabilised thanks to the recruitment of the ubiquitin hydrolase USP14 which removes K48-linked ubiquitin chains. The role of IP in the regulation of the signalling events leading to type I IFN responses is poorly understood. It has been shown that IP deficiency resulted in decreased phosphorylation of IRF3 in response to LPS, although the underlying mechanisms remain obscure [96]. In line with this, data from our group suggest that microglia lacking β5i result in the stabilisation of various key components of the signalling pathways engaged by LPS (unpublished observations).

## 3. The UPS as a Major Guardian of Proteostasis during Activation of the Immune System

Proteostasis is a mechanism of cellular homeostasis ensuring the sensitive balance between synthesis, folding, trafficking, and degradation of proteins [97,98]. In the protein life cycle, trafficking is of particular importance, because all proteins are sorted into their distinct destinations to ensure their correct function. One major sorting pathway in the cell is the cytosolic pathway including nuclear, peroxisomal, mitochondrial, and cytosolic proteins. However, about 30% up to 60% of all proteins follow the secretory pathway and are sorted into ER, Golgi, lysosomal, plasma membrane or extracellular compartments depending on cell function [99]. Because immune cells massively use this pathway for the production of humoral immune factors such as cytokines or immunoglobulins, they are particularly sensitive to perturbations in protein transport. Thus, immune cell function heavily relies on proteostatic mechanisms to avoid unbalanced immune responses or immune cell death [100,101].

Proteostasis is highly challenged by ageing and many pathological situations such as infection, inflammation or oxidative stress that result in protein damage and/or misfolding [102,103,104]. Accumulation of protein aggregates causes cellular proteotoxic stress that triggers several cellular responses for adaptation including the unfolded protein response (UPR), the integrated stress response, and immune responses (ISR) as well as the upregulation of alternative proteasome isoforms to rebalance the system for adaptation [105]. Such proteostasis imbalances cause specific pathologies, so-called proteinopathies, that include neurodegenerative diseases characterized by the accumulation of ubiquitin–protein conjugates [106,107]. The proteostasis network thus represents adaptation mechanisms to increased proteotoxic burden and enables a functional proteome to maintain the health of the living cell.

Both the UPR and ISR have evolved as essential protein quality control (PQC) systems to counteract disrupted proteostasis and prevent the development of proteinopathies. These programs sense proteostatic imbalances by receptors in the cytosol (for the ISR) and the ER (for the UPR) and initiate the production of rescue factors as well as a global translational arrest. As illustrated in Figure 4, the phosphorylation of the translation initiation factor eIF2α by the two cytosolic kinases general control non-derepressible 2 (GCN2) [108], and double-stranded RNA-dependent protein kinase (PKR) [109] or protein kinase RNA-like endoplasmic reticulum kinase (PERK) [110] is a central event at the intersection of these pathways.

All eIF2α kinases share similar kinase catalytic domains, dimerize, and autophosphorylate for full activation. Reflecting their unique regulatory mechanisms, each kinase responds to distinct environmental and physiological stimuli. While the heme-regulated inhibitor (HRI) is mainly expressed in erythroid cells to synchronize the availability of heme and iron with globin translation and the production of haemoglobin, all other kinases are ubiquitously expressed [111]. GCN2 is activated by metabolic stimuli such as amino acid starvation to coordinate translation with the availability of amino acids. Interestingly, it has been shown that proteasome inhibition also promotes the activation of GCN2 due to missing amino acid delivery from protein decay and recycling mechanisms [112,113,114]. PKR appears to be the most versatile kinase in terms of inducers. Although double-stranded RNA and virus infection represent the main activating agents for PKR, it has been shown that oxidative, ER-, or ribotoxic stress, growth factor deprivation, cytokines, or stress granules activate PKR as well [115]. By contrast, PERK is activated by misprocessed ER proteins in the lumen or membrane to catalyse eIF2α phosphorylation and the concomitant downstream events [110,116]. The global translational arrest upon eIF2α phosphorylation results in focused translation of ATF4 and other transcription factors. Phosphorylation of eIF2α can be counteracted by eIF2α phosphatases such as the protein phosphatase 1 (PP1) complex that recruits a PP1 catalytic subunit (PP1c) and one of the two regulatory subunits PPP1R15A or GADD34 [117,118]. As shown in Figure 4, perturbations of protein homeostasis in the ER lumen are sensed by three ER-transmembrane receptors: inositol-requiring protein 1α (IRE-1α), activating transcription factor 6 (ATF6), and PERK. Under physiological conditions, IRE-1α, ATF6, and PERK are associated with the molecular chaperone binding immunoglobulin protein BiP/GRP78 and reside in the inactive form [119,120]. In response to ER stress, BiP/GRP78 disassociates from the IRE-1α, PERK, and ATF6 to activate the three branches of the UPR. In addition to activating the splicing of the transcription factor Xbp1 to sXBP1, IRE-1 has been linked to the degradation of various cytosolic mRNAs, a process known as regulated IRE-1-dependent decay (RIDD) [121,122]. ATF6 is an ER membrane-bound transcription factor itself that is processed by the Golgi proteases S1P and S2P to become activated [123,124]. Transcription factors induced by all branches of the UPR and/or the ISR trigger the upregulation of stress genes. These genes encode rescue factors such as chaperones to increase the protein folding capacity, components of the ER-associated degradation (ERAD) for removal of irreversibly damaged proteins, and factors that control amino acid and lipid metabolism, cell death, or autophagy (Figure 4) [118,125,126]. Thus, reprogramming of the proteostasis network by downregulation of the global protein synthesis is a common downstream process of ISR and UPR.

Another adaptation mechanism to combat proteotoxic stress represents the upregulation of proteasome isoforms upon cytokine signalling or infection. IP are in particular important for rapid clearance of damaged or misfolded proteins to sustain cellular protein homeostasis under interferon-induced oxidative stress [27]. Unlike SP, IP have a high affinity for the interferon-γ (IFN-γ)-inducible PA28α/β regulatory particle [127]. As a response to cellular stress, e.g., oxidative stress, IP can assemble with both 19S and PA28α/β regulators to form a hybrid proteasome (HP, 19S-20S-PA28α/β) [128,129], thereby enhancing the rapid degradation of oxidant-damaged proteins in a ubiquitin-dependent manner. As such, it prevents the accumulation of protein aggregates and produces antigenic peptides for MHC class I antigen presentation [130,131]. Cytokine or TLR signalling causes an increase in ubiquitin–protein conjugates in different immune cells [68,132,133]. Importantly, IP formation prevents the accumulation of ubiquitin-modified proteins, as evidenced by increased aggregation of ubiquitin-positive inclusions in IP-lacking or IP specific ONX-0914 inhibitor-treated [27,69,134,135]. Proteostasis disruption caused by UPS impairment is especially challenging for myeloid cells. Our recent study revealed that inhibition of proteasomes (both SP and IP) by bortezomib results in accumulation of ubiquitin–protein conjugates and subsequent activation of the UPR, thereby resulting in the generation of a type I IFN response in myeloid cells [135]. Specifically, the activation of the IRE1α-dependent arm of UPR was mainly responsible for the induction of type I IFN signalling in microglia as well as the human monocytic cell line THP-1 following proteasome impairment. As alluded to earlier, the accumulation of proteins following disrupted protein homeostasis and neuroinflammation is a very common feature of most of the neurodegenerative diseases such as Alzheimer’s Disease (AD), Parkinson’s Disease (PD), and Huntington Disease (HD) [106,136,137]. Thibaudeau et al. have shown that toxic oligomers, e.g., amyloid-beta (Aβ), alpha-synuclein (α-syn), and mutant Huntington (Htt), which share a common 3D structure, inhibit proteasome activity and thereby sustain protein aggregation in vitro [138]. Nevertheless, further investigation of the UPS-mediated proteostasis in myeloid cells is needed to better understand the pathogenesis behind the proteasome impairment and its contribution to the development of neurodegenerative diseases.

## 4. Role of the UPS in MHC Class I Antigen Processing and Presentation

Antigen presentation is a key process for the initiation of primary adaptive immune responses, in the course of which foreign antigens in the form of short peptide fragments are presented on the surface of cells to cytotoxic T cells (CTL). Antigens are presented by two main classes of major histocompatibility complexes (MHCs). MHC class II molecules are expressed exclusively on the surface of professional antigen-presenting cells to present extracellular foreign antigens, which are processed in the lysosome (e.g., from phagocytosed bacteria or viruses) [139,140]. MHC class I molecules are expressed by all nucleated cells and bind self-peptides derived from cellular proteins as well as foreign peptides derived from proteins arising from intracellular pathogens such as viruses. Peptides to be presented by MHC class I molecules are typically 8–10 amino acids in length and originate from proteasome-dependent protein breakdown. Although both SP and IP are able to generate peptides for MHC class I presentation, the incorporation of the inducible subunits into proteasomes may be either beneficial or detrimental in this process. Hence, a substantial number of studies highlighted a positive role of IP in the initiation of CTL responses directed against viral and bacterial antigens [141,142,143,144,145,146]. However, IP have also been reported to abrogate the presentation of tumour epitopes [147,148,149,150]. The differences observed between SP and IP in the supply of MHC class I-restricted peptides may be easily explained by the fact that IP subunits exhibit a higher cleavage rate than their standard counterparts [151] which, depending on peptide sequence, may result in increased epitope generation or destruction.

In addition, both SP and IP are also capable of generating neoepitopes composed of two different fragments from peptide sequences which are not contiguous in the parent antigen [152,153,154,155,156,157,158,159]. Such peptides emerge from the fusion of proteasomal cleavage products in a process referred to as proteasome catalysed peptide splicing (PCPS) [160,161]. Because of the technical difficulties to detect such spliced peptides, the contribution of PCPS to the MHC class I peptidome has long been underestimated or even ignored [162]. However, a study reporting that one third of cell surface MHC class I ligands are represented by PCPS-generated peptides has recently brought PCPS into focus of immunology research [163]. As shown in Figure 4, proteasome-generated peptides are subsequently transported from the cytosol into the ER via transporter associated with antigen processing (TAP) and loaded onto the MHC class I molecules bound to the ER membrane [164,165,166,167,168]. Peptide-loaded MHC class I molecules in turn dissociate from the ER and are translocated to the cell surface for presentation to CTL which can directly eliminate infected cells [169]. Moreover, dendritic cells and macrophages can also present extracellular antigens by cross-presentation on MHC class I molecules in proteasome-dependent [43,170,171,172,173,174] and -independent [175,176,177,178,179,180] mechanisms. As initially postulated by Yewdell and colleagues [181], the major peptide source for MHC class I antigen presentation derives from defective ribosomal products (DRIPs), which appear due to errors during protein synthesis [182,183]. DRIPs are immediately ubiquitinated polypeptides at the ribosome and are degraded by proteasomes in situ. The usage of DRIPs as antigenic source ensures the presentation of peptides derived from almost all proteins irrespective of their localization [184,185,186,187,188,189,190]. Protein synthesis and thus, DRIP formation, is induced by infection or inflammation triggered by IFN signalling [27,132,191,192]. Enhanced degradation capacities of IP assure efficient generation of peptides for MHC class I antigen presentation as shown in IP-deficient mice [142,193]. Thus, the IP function relies on the clearance of inflammation-induced DRIPs and inflammation-damaged proteins, thereby improving peptide supply for antigen presentation [27].

## 5. The UPS in Disease

### 5.1. Autoinflammatory Diseases

As described above, proteasome dysfunction has been mainly correlated with proteinopathies of the central nervous system. Therefore, it was somehow unexpected that genetic alterations (e.g., loss of function mutations) in genes encoding proteasome components or their assembly factors have been found in patients suffering from autoinflammation. Autoinflammatory disorders represent sterile inflammatory conditions characterized by episodes of early-onset fever and disease-specific patterns of organ inflammation including neuroinflammation [194] called CANDLE/PRAAS (Chronic Atypical Neutrophilic Dermatosis with Lipodystrophy and Elevated Temperature/Proteasome Associated Autoinflammatory Syndromes) [195,196]. Depending on the laboratory in which such alterations were characterized, these syndromes have also been referred to as joint contractures, muscle atrophy, microcytic anaemia, and panniculitis-induced lipodystrophy (JMP), Nakajo-Nishimura syndrome (NKJO), POMP-related autoinflammation and immune dysregulation disease (PRAID), and CANDLE/PRAAS [197,198,199,200,201]. As shown in Table 1, mutations in the *PSMB8* gene, encoding the IP subunit β5i/LMP7, were the first identified to cause CANDLE/PRAAS [197,199,200]. Therefore, it was proposed that CANDLE/PRAAS might be an IP deficiency disease. However, in 2015 Brehm et al. identified other mutations in genes coding for IP as well as SP 20S subunits: *PSMA3* (encodes proteasome subunit α7), *PSMB4* (encodes proteasome subunit β7), *PSMB9* (encodes proteasome subunit β1i/LMP2), as well as *POMP* (encodes proteasome maturation protein, POMP) [201].

A couple of years later, further mutations were identified in *POMP* [202], in the proteasome assembly chaperone 2 gene, *PSMG2* [203], as well as in the third IP gene *PSMB10* encoding the β2i/MECL1 subunit [204]. Regardless of the genomic alteration and subunit, the major molecular consequence of alterations in cells of CANDLE/PRAAS patients is a decreased proteasome assembly and activity of proteasomes leading to an accumulation of ubiquitinated proteins [198,199,201,202,203,204]. A common clinical feature of CANDLE/PRAAS is the systemic inflammation including neuroinflammation. Consistent with the phenotype observed in cells upon proteasome impairment, CANDLE/PRAAS patients are characterized by a chronic secretion of pro-inflammatory cytokines and type I IFN [196,201,203], thereby placing these conditions into the category of interferonopathies [194]. Intriguingly, proteasome loss-of-function mutations are not always associated with systemic autoinflammation. Herein, subjects with alterations in *PSMD12*, *PSMC3* or *PSMB1* suffer from neurodevelopmental disorders rather than typical CANDLE/PRAAS syndromes [205,206,207,208]. The reason for these varying phenotypes is still unclear and warrants further investigation.

Apart from proteasome genes, further components of the UPS may also play a role in the activation of inflammatory signalling pathways, when altered. For example, mutations in A20, an enzyme with dual DUB/E3 function [81,209], has been shown to cause dysregulation of NF-κB signalling in patients suffering autoinflammatory disorders (AIDs) [210,211]. Further examples of UPS alterations outside proteasome genes include mutations in the E3 ubiquitin ligase *Itch*, which results in truncation of the ITCH protein and ITCH deficiency. They are identified as a cause of syndromic multisystem autoimmune disease whose major phenotype is chronic lung disease [212]. Importantly, *Itch* has been shown to cooperate with other E3 ligases including WWP2 and Cbl-b [213,214] and its deficiency may compromise the breakdown of multiples targets in immune cells [215,216,217,218,219,220,221].

### 5.2. Manipulation of the UPS by Pathogens

In the above chapters, the UPS was outlined as an important player in the regulation of immune cell function and immune responses. Therefore, it is not very surprising that many pathogens have evolved sophisticated mechanisms to manipulate the UPS to ensure immune escape and survival in the host. In this chapter, two distinct examples of pathogenic mechanisms of UPS manipulation will be explained in more detail.

In 1976 an outbreak of an unknown disease occurred during a convention of war veterans in Philadelphia, USA. The causative agent was later identified as the gram-negative bacterium *Legionella pneumophila* causing severe lung inflammation known as Legionnaire’s diseases which are typically caught following inhalation of infected droplets and/or water from air conditioning systems. *L. pneumophila* provides an interesting example of pathogens capable of evading the immune system by hijacking the host UPS. Herein, *L. pneumophila* has been shown to exploit host proteasomes by expressing ubiquitin ligases that target other bacterial proteins for degradation when these are no longer required. One of these E3 ligases encoded by *L. pneumophila* include the U-box protein LubX which mediates the ubiquitination and subsequent proteasome-mediated degradation of its own effector protein SidH which is beneficial only during the early phase of infection [222,223,224].

In addition to influencing protein breakdown, *L. pneumophila* also impacts vesicle trafficking through the ubiquitination of the host ER-associated Rab small GTPases Rab33b and Rab1 by the ubiquitin ligase SidE [225,226]. Strikingly, the ubiquitin-conjugation system initiated by SidE differs fundamentally from the traditional one employed by eukaryotic cells. Indeed, instead of using ATP and the canonical three-enzyme cascade, it uses NAD to activate ubiquitin and form an ADP-ribose-ubiquitin intermediate whose ubiquitin moiety can then be directly transferred to intracellular substrates in an E1/E2-independent manner [226,227]. In addition, the phosphoribosyl moiety of the ADP-ribose-ubiquitin intermediate can also be used to modify further latent ubiquitin molecules so that these cannot be used for conjugation anymore, thereby impairing critical cellular processes of the innate immune response, inducing protein breakdown and signal transduction [227]. Besides SidE, another “all-in-one” ubiquitin ligase of *L. pneumophila* includes the transglutaminase MavC, which attaches ubiquitin via glutamine (Gln40) to UBE2N using two possible target lysine’s (Lys92 and Lys94) [228]. Mono-ubiquitination of UBE2N by MavC results in its inactivation, thereby abolishing the formation of K63-polyubiquitin chains required for NF-κB signalling. Importantly, *L. pneumophila* also expresses a significant number of DUBs which counteract the ubiquitination of intracellular host proteins [229]. These include SdeA that damps NF-κB signalling by preferentially trimming ubiquitin K63-linkages [230].

Since the recent outbreak of severe acute respiratory syndrome coronavirus 2 (SARS-CoV-2) β-coronaviruses are back in the focus of the scientific community. Similar to many other viruses, β-coronaviruses interfere with the host UPS to evade immune recognition and ensure viral replication. Of particular interest is the viral papain-like protease (PLpro), which acts as a DUB with broad specificity in SARS-CoV-1 and -2 as well as in the *Middle East respiratory syndrome-related coronavirus* (MERS-CoV) [231,232,233,234]. By removing the K48-linked polyubiquitin chains from IκBα, PLpro prevents IκBα proteasomal degradation and therefore dampens NF-κB activation [235]. Interestingly, beside trimming ubiquitin chains, PLpro is capable of removing the ubiquitin-like modifier ISG15 from intracellular proteins [231]. In this regard, the de-ISGylation of IRF3 by PLpro has been shown to destabilize IRF3 and thus contribute to the evasion of coronaviruses to type I IFN responses [236].

### 5.3. Tumour Diseases

As previously stated, immune cell function strongly relies on a balanced proteostasis and a functional UPS. Thus, it is not surprising that the complete hematopoietic system and stem cell function in general are dependent on the timely degradation of signalling molecules during haematopoiesis and differentiation [237,238,239,240]. Consequently, the UPS and the proteostatic potential of hematopoietic cells represent targets of cancer therapy in hematopoietic malignancies [241]. Leukaemia is a group of blood cancers originating from transformation of haematopoietic stem or progenitor cells. Many leukaemia treatment strategies rely on the generation of genotoxic stress in highly proliferating cancer cells [242]. We will focus on two alternative strategies targeting components of the UPS, e.g., the proteasome or the cereblon (CRBN) E3-ubiquitin ligase.

Since its approval in 2003, pharmacological inhibition of proteasome activity by small molecules such as bortezomib (BTZ), also known under the brand name Velcade^®^, has become the first line therapy of multiple myeloma and mantle cell lymphoma. This led to significant improvement of survival rates of myeloma patients [243]. Meanwhile, because of side effects and BTZ resistance, a couple of additional proteasome inhibitors (Carfilzomib, Ixazomib, Delanzomib, Oprozomib, and Marizomib) with broader anti-cancer activities and altered pharmacological properties have been approved for clinical use. These also show improved clinical outcome in myeloma patients who relapsed upon initial BTZ treatment [242,244,245,246,247]. Thanks to the good efficacy of proteasome inhibitors in multiple myeloma or mantle cell lymphoma, several clinical trials have been launched to test responsiveness of other leukaemia types to proteasome inhibitors. Molecular mechanisms of BTZ resistance are not fully understood; however, three major features are observed in patients with relapses, namely, (i) genetic alterations in β5, (ii) altered proteasome subunit compositions, and (iii) induction of proteasome subunits. Genetic analysis of BTZ-resistant cells revealed a hot spot for mutations in exon 2 of *PSMB5* encoding the β5 subunit. All mutations affect the BTZ binding site of β5, alanine in position 49, directly or in close proximity to, and prevent efficient binding of BTZ [242]. Another resistance mechanism relies on alteration of immuno-/constitutive proteasome ratio either by up- or down-regulation of immunoproteasomes [248]. Upregulation of the β5 subunit partly accompanied by the induction of other subunits represents the third resistance mechanism.

Proteasome abundance in cells is regulated by the transcription factor TCF11/Nrf1 encoded by the *NFE2L1* gene [249]. TCF11/Nrf1 is an ER-membrane-resident protein, which is ubiquitin-modified by the E3 ligase Hrd1 and permanently degraded by ERAD under non-stressed conditions. Upon proteotoxic stress triggered by proteasome inhibitors, TCF11/Nrf1 is deglycosylated by NGLY1, pulled out of the ER-membrane by the AAA-ATPase p97 and cleaved by the aspartyl protease DDI2 (DNA-damage inducible 1 homolog 2). The cleaved form of TCF11/Nrf1 is translocated into the nucleus, where it binds to antioxidant response elements (AREs) in the promoter regions of almost all proteasome genes, and induces their expression [250,251,252] in a process referred to as bounce back response [253]. The complete NGLY1-DDI2-TCF11/Nrf1 axis is discussed as a new drug target for myeloproliferative neoplasms [254,255,256].

Thalidomide, also known under its brand name Contergan, initiated one of the biggest pharmacological disasters and scandals in Germany. More than 50 years ago, this drug was advertised as a harmless sleeping and tranquillizing agent that helps against morning sickness in pregnancy, but caused embryonic damage to thousands of children because of its teratogenic potential [257,258]. Half a decade later, thalidomide or its derivatives lenalidomide, and pomalidomide experience a revival as anticancer drugs for treatment of hematologic malignancies. The direct target of such thalidomide-like drugs is the protein cereblon, which together with the damaged DNA binding protein 1 (DBB1), cullin-4A (CUL4A), and regulator of cullins 1 (ROC1) forms an E3 ubiquitin ligase complex [259]. This ubiquitin ligase complex controls pathways involved in limb outgrowth of embryos and other developmental pathways [257]. Binding of thalidomide to the substrate recognition protein cereblon alters the substrate portfolio of the cereblon containing the E3 ubiquitin ligase complex. It has been demonstrated that the zinc finger-containing transcription factors Ikaros (IKZF 1) and Aiolos (IKZF3) are selectively bound by cereblon upon thalidomide treatment and target these transcription factors for degradation. Consequently, this causes cytotoxicity to multiple myeloma cells. In addition, Ikaros and Aiolos control cytokine production. Therefore, thalidomide and its derivatives also exert immunomodulatory effects [258,260,261]. In conclusion, components of the UPS turned out to be very important drug targets in the fight against cancer, and the future certainly will bring up some more promising candidates.

## Figures and Tables

**Figure 1 biomolecules-11-00060-f001:**
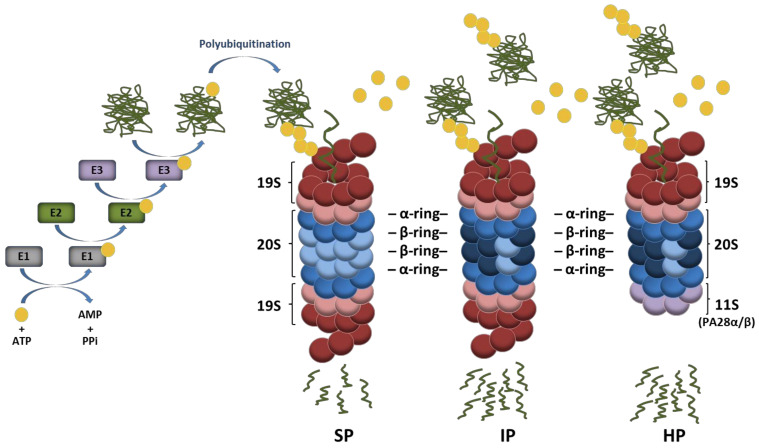
Ubiquitin conjugation pathway of protein substrates and their subsequent degradation by proteasome complexes. Ubiquitin activation requires its transfer to an E1 ubiquitin-activating enzyme under ATP hydrolysis. In a second step, the activated ubiquitin is transferred to an E2 ubiquitin-conjugating enzyme and used for the covalent modification of intracellular proteins with the support of E3 ubiquitin ligases. Protein substrates may be marked with one single ubiquitin molecule on one acceptor site resulting in mono-ubiquitination. The ubiquitin molecule may also be subjected to ubiquitination, thereby leading to the formation of various polyubiquitin chains, among which the K48-linkages represent the most prevalent proteasome-targeting signals. Major proteasome complexes include standard proteasomes (SP), immunoproteasomes (IP), and hybrid proteasomes (HP) which distinguish themselves by the nature of their catalytic subunits and/or regulators (19S, 11S). It is understood that both IP and HP exhibit a higher capacity of degrading ubiquitin-modified proteins than SP.

**Figure 2 biomolecules-11-00060-f002:**
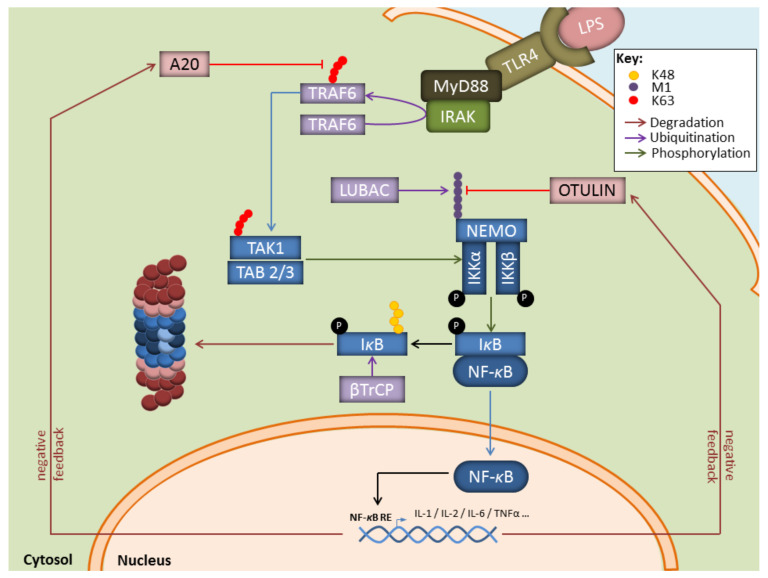
Key contributions of the UPS to the regulation of the NF-κB signaling pathway in response to LPS. Depicted is the NF-B pathway emanating from the cell surface receptor TLR4 upon LPS binding. NF-κB signalling relies on the upstream E3 ubiquitin ligase TRAF6 which catalyzes K63-linked polyubiquitin chains which themselves activate the TAK kinase complex. Activated TAK1 phosphorylates the downstream heterotrimeric IKK kinase complex whose full activation further requires the generation of Met1-linked linear polyubiquitin chains by LUBAC. Once activated, IKK promotes the phosphorylation of κBα and its subsequent proteasomal degradation following its K48-ubiquitination by SCF*^TRCP^*, thereby allowing NF-κB nuclear translocation. Genes induced by NF-κB include those coding for deubiquitinating enzymes Otulin and A20 which ensure a negative feedback loop by trimming linear and K63-polyubiquitin chains.

**Figure 3 biomolecules-11-00060-f003:**
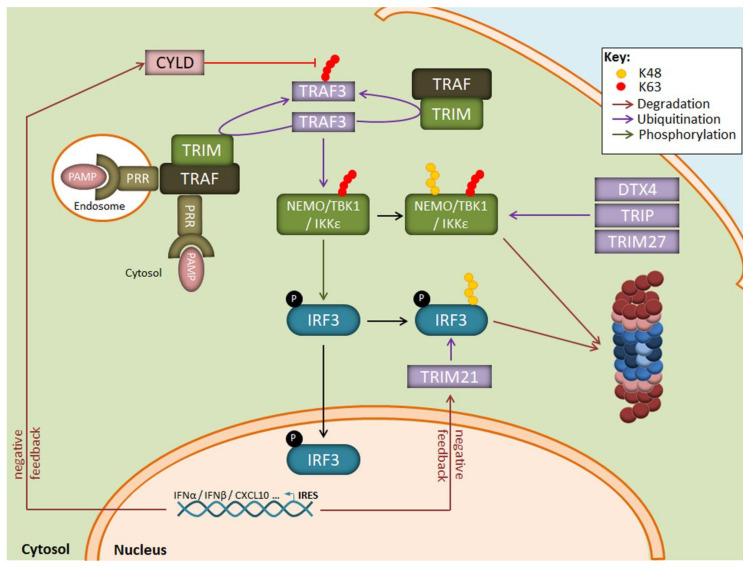
Regulation of the IRF3 signaling pathway by the UPS following LPS exposure. Binding of PAMP to intracellular PRR results in the autoactivation of the E3 ubiquitin ligase TRAF3 which activates the downstream TBK1/IKK1ε kinase complex by generating K63-linked polyubiquitin chains. The TBK1/IKK1ε complex catalyzes the phosphorylation of IRF3, thereby inducing its nuclear translocation and the transcription of genes coding for type I IFN. A prolonged activation of TBK1/IKK1ε by TRAF3 is prevented by the E3 ubiquitin ligases DTX4, TRIP, and TRIM27 which modify TBK1/IKK1ε with K48-linked polyubiquitin chains and target it for proteasomal degradation. Negative feedback mechanisms also include the upregulation of the CYLD and TRIM21 proteins which remove K63-linkages and target IRF3 for proteasomal degradation, respectively.

**Figure 4 biomolecules-11-00060-f004:**
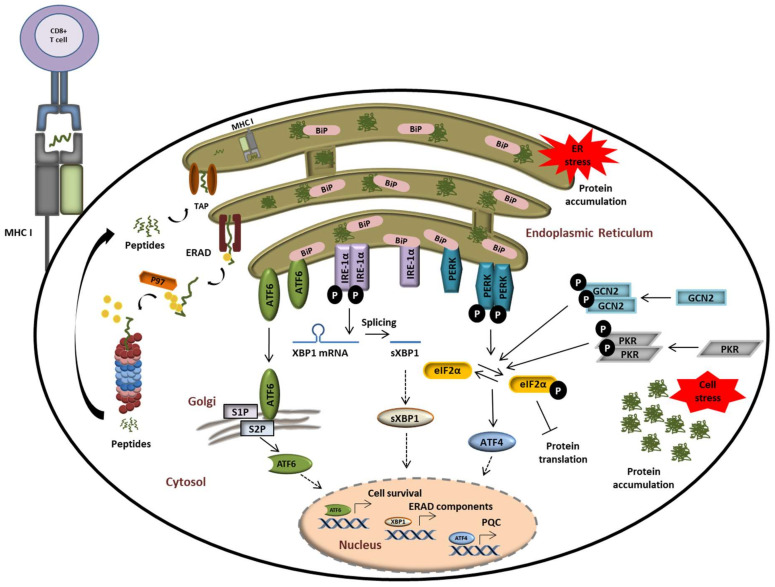
The UPS plays a central role in alerting and resolving imbalances of protein homeostasis. Proteostasis disruption is typically characterized by the accumulation of misfolded and/or damaged proteins in the ER lumen which fail to be retro-translocated to the cytosol by ERAD for proteasomal degradation. Protein aggregation in the ER is sensed by the three ER receptors IRE1α, ATF6, and PERK which then undergo activation to trigger the UPR. This ultimately results in the activation of transcription factors ATF6, sXBP1, and ATF4, thereby promoting the transcription of genes coding for proteins destined to compensate the burden of unfolded/damaged proteins. Perturbed protein homeostasis is also sensed in the cytosol by GCN2 and PKR which undergo autoactivation following amino acid deficiency and stress granules, respectively. Both of these kinases phosphorylate eIF2α, thereby intersecting with the PERK-mediated pathway of the UPR.

**Table 1 biomolecules-11-00060-t001:** List of disorders associated with genetic alterations of genes coding for UPS components.

Disease	Altered UPS Gene	Component
CANDLE/PRAAS	*PSMB8* (β5i/LMP7)	20S
	*PSMA3* (α7)	20S
	*PSMB4* (β7)	20S
	*PSMB9* (β1i/LMP2)	20S
	*POMP* (POMP)	Proteasome assembly
	*PSMG2* (PAC2)	Proteasome assembly
	*PSMB10* (β2i/MECL-1)	20S
Neurodevelopmental Disorders	*PSMC3* (Rpt5)	19S
	*PSMD12* (Rpn5)	19S
	*PSMB1* (β6)	20S
A20 Haploinsufficiency	*TNFAIP3* (A20)	DUB
Syndromic Multisystem Autoimmune Disease	*ITCH* (ITCH)	E3 ligase

## Data Availability

Not applicable.

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
