# Peer review of "The Ubiquitin–Proteasome System in Immune Cells"

_biomolecules, 2021, doi:10.3390/biom11010060_

Round 1
Reviewer 1 Report
Although clearly written and often informative, it is sometimes a bit difficult to understand the exact topic of this review. The title might be ambiguous but in the abstract and the introduction the authors clearly define the UPS as exclusively associated with protein degradation, especially when it is mentioned lines 60-62 that: “… while those linked via K63 allow the clearance of damaged organelles (i.e., ribosomes, endoplasmic reticulum and/or mitochondria) by the autophagy-lysosome pathway [17].”. Therefore, the reader may find the following sections (or at least large parts of these sections) a bit confusing since they deal with events unrelated to protein degradation. An effort should be made to focus more clearly on UPS-related matters and to help the non-expert reader distinguishing Ub events triggering degradation from others.
For instance:
- Section 2 deals mostly with events not related to degradation, especially K63-related ones (revealing that K63 ubiquitination does not only allow the clearance of damaged organelles as stated before). Maybe just strengthening at the beginning of this section, and for the rest of the review, that ubiquitination processes resulting in degradation are often intricately coupled with other ubiquitination processes regulating protein activities/interactions and indicating that K63 ubiquitination plays a major part in this, would allow to smoothly introduce the genuine UPS-related events (among them IkB degradation) that are the main topics of this review. (More specific comments about this section below).
- In Section 5.1 (lines 381-400), it is also difficult to understand why several disorders (Otulipenia, A20 haploinsufficiency and Syndrome Multisystem Autoimmune disease) are mentioned as related to UPS dysfunction especially if, as explained by the authors, they are characterized by autoimmunity/inflammation and mutated components are DUBs negatively regulating ubiquitination steps unrelated to degradation. For sure, A20 might represent an exception since it may exhibit dual DUB/E3 ligase activities controlling K48-linked ubiquitination and Itch may directly or indirectly be involved in degradation events regulated by the UPS but this is not even mentioned.
- In Section 5.2, in the L. pneumophila part, SidE influences vesicle trafficking and MavC acts on K63-linked ubiquitination not linked to degradation. Is there anything firmly linked to dysfunction of the UPS? Lines 429 to 438, the part dealing with TB is also not very informative regarding a regulation of the UPS by this specific pathogen (Even the authors admit poorly characterized links: “Although the precise role of these UPS genes during Mtb infection remains to be fully determined, one cannot exclude that they impact Mtb ubiquitination. »).
Specific comments:
The two Figs 2 and 3 are similar in presentation and, because of that, are misleading. In Fig. 2, the specific pathway induced by TLR4 after binding of LPS is rightly represented as dependent on the localization of TLR4 at the cytoplasmic membrane. In Fig. 3, which describes the activation of IRF3 by TLRs different from TLR4 (recognizing various PAMPs but not LPS) and RIG-I-like receptors, these receptors should be intracellularly located. The text should be modified accordingly, lines 169-171, by saying something like “These include notably K63-linked polyubiquitin chains catalyzed by the E3 ubiquitin ligase TRAF3 following recognition of viral products by some intracellular TLRs, such as TLR3, TLR7, TLR8 or TLR9, and/or RIG-I-like receptors [90-92].” Moreover, in the text a NEMO/TBK1/IKKepsilon is mentioned but does not appear in Fig. 3. Finally, legend of Fig. 3 should be modified by removing the mention to LPS, which is the specific trigger of TLR4 signaling presented in Fig.2.
Title of Section 5.2 should be : “Manipulation of the UPS by Pathogens” instead of “Manipulation of the UPS in by Pathogens”
Author Response
Reviewer #1
Comments and Suggestions for Authors
Although clearly written and often informative, it is sometimes a bit difficult to understand the exact topic of this review. The title might be ambiguous but in the abstract and the introduction the authors clearly define the UPS as exclusively associated with protein degradation, especially when it is mentioned lines 60-62 that: “… while those linked via K63 allow the clearance of damaged organelles (i.e., ribosomes, endoplasmic reticulum and/or mitochondria) by the autophagy-lysosome pathway [17].”. Therefore, the reader may find the following sections (or at least large parts of these sections) a bit confusing since they deal with events unrelated to protein degradation. An effort should be made to focus more clearly on UPS-related matters and to help the non-expert reader distinguishing Ub events triggering degradation from others.
For instance:
- Section 2 deals mostly with events not related to degradation, especially K63-related ones (revealing that K63 ubiquitination does not only allow the clearance of damaged organelles as stated before). Maybe just strengthening at the beginning of this section, and for the rest of the review, that ubiquitination processes resulting in degradation are often intricately coupled with other ubiquitination processes regulating protein activities/interactions and indicating that K63 ubiquitination plays a major part in this, would allow to smoothly introduce the genuine UPS-related events (among them IkB degradation) that are the main topics of this review. (More specific comments about this section below).
>> The referee raises a valid point here and we agree with him/her that the main emphasis of this review is the ubiquitin-proteasome system (UPS) and not ubiquitination per se. Even if both of these processes are not dissociable, we acknowledge the fact that too much detail on non-proteolytic ubiquitin chains may confuse the reader. As such, we followed his/her excellent suggestion and removed most of the parts addressing the role of K63- and Met1-linkages in the signalling pathways triggered by LPS and viruses (page 5, lines 134-149). As requested by the reviewer, we nonetheless mentioned at the beginning of this section that UPS-driven protein breakdown in signal transduction is intrinsically associated with the generation of non-proteolytic chains that activate key signalling components (page 5, lines 131-133). We believe that this part of the manuscript is now much more focussed on the roles of K48-linked ubiquitin chains and proteasomal degradation in the regulation of cell signalling initiated by pathogen recognition receptors.
- In Section 5.1 (lines 381-400), it is also difficult to understand why several disorders (Otulipenia, A20 haploinsufficiency and Syndrome Multisystem Autoimmune disease) are mentioned as related to UPS dysfunction especially if, as explained by the authors, they are characterized by autoimmunity/inflammation and mutated components are DUBs negatively regulating ubiquitination steps unrelated to degradation. For sure, A20 might represent an exception since it may exhibit dual DUB/E3 ligase activities controlling K48-linked ubiquitination and Itch may directly or indirectly be involved in degradation events regulated by the UPS but this is not even mentioned.
>> We fully agree with the reviewer that placing the genes Otulin, ISG15 and USP43 into our list of autoinflammatory disorders associated with UPS alterations may be confusing to the reader, since these components are not intrinsically involved in UPS-driven protein breakdown which is the central topic of this review. In an attempt to remain focussed on the relevance of the UPS in immune cells, we followed his/her suggestion by removing them from Table 1 (page 12). Accordingly, we also removed parts of text in this section referring to these genes in the revised version of our manuscript (page 13, lines 410-430). We also agree with the reviewer that we failed to mention the ability of A20 and Itch to act as E3 ubiquitin ligases and target substrates for proteasome-mediated degradation. This point is now clarified in the revised version of our manuscript (page 13, lines 409-410 and 418-420).
- In Section 5.2, in the L. pneumophila part, SidE influences vesicle trafficking and MavC acts on K63-linked ubiquitination not linked to degradation. Is there anything firmly linked to dysfunction of the UPS? Lines 429 to 438, the part dealing with TB is also not very informative regarding a regulation of the UPS by this specific pathogen (Even the authors admit poorly characterized links: “Although the precise role of these UPS genes during Mtb infection remains to be fully determined, one cannot exclude that they impact Mtb ubiquitination. »).
>> As perfectly pointed out by the referee, the UPS hijacking by Legionella described in this section exclusively refers to non-proteolytic processes (i.e. vesicle trafficking and ubiquitin-trimming). We followed his/her advice and added a paragraph to this section describing an interesting mechanism by which Legionella hijacks host proteasomes (page 14, lines 443-447). In this process, Legionella is capable of targeting its own effector proteins for proteasome-mediated degradation thanks to the expression of a U-box ubiquitin E3 ligase named LubX. We also agree with the reviewer that the ability of Tuberculosis (TB) to manipulate the host UPS is not convincingly addressed in this section. We believe this lack of clarity reflects the failure of the current literature to describe robust and precise mechanism in this regard. Because of its speculative nature, we followed the referee’s suggestion and decided to remove this part from the manuscript.
Specific comments:
The two Figs 2 and 3 are similar in presentation and, because of that, are misleading. In Fig. 2, the specific pathway induced by TLR4 after binding of LPS is rightly represented as dependent on the localization of TLR4 at the cytoplasmic membrane. In Fig. 3, which describes the activation of IRF3 by TLRs different from TLR4 (recognizing various PAMPs but not LPS) and RIG-I-like receptors, these receptors should be intracellularly located. The text should be modified accordingly, lines 169-171, by saying something like “These include notably K63-linked polyubiquitin chains catalyzed by the E3 ubiquitin ligase TRAF3 following recognition of viral products by some intracellular TLRs, such as TLR3, TLR7, TLR8 or TLR9, and/or RIG-I-like receptors [90-92].” Moreover, in the text a NEMO/TBK1/IKKepsilon is mentioned but does not appear in Fig. 3. Finally, legend of Fig. 3 should be modified by removing the mention to LPS, which is the specific trigger of TLR4 signaling presented in Fig.2.
>> We are grateful to the referee for pointing out these mistakes. We corrected Fig. 3 accordingly by (ii) removing the cell surface pathogen recognition receptor (PRR), (ii) including both cytosolic and endosomal PRR and (iii) illustrating the NEMO/TBK1/IKKε complex. As requested by the reviewer, we also modified the text related to this figure and indicated that PRR recognizing viral products may be intracellular Toll-like receptors (TLR) and/or RIG-like-I receptors (page 6, lines 181-182). We finally removed the mention to LPS in the figure legend (page 6, line 186).
Title of Section 5.2 should be : “Manipulation of the UPS by Pathogens” instead of “Manipulation of the UPS in by Pathogens”
>> This has been changed.
Reviewer 2 Report
This is an excellent review, valuable summary on recent progress on the role of UPS in immune system. It has a logical structure, easy to follow explanations and comprehensive original figures. Chapter 5 about the contribution of UPS to certain diseases contains even a paragraph on the actual SARS-CoV-2.
A suggest a few minor corrections:
line 28: instead "ubiquitination is mediated by three groups of enzymes" write "ubiquitination is mediated by an enzyme cascade"
line 38: "via oxyester or thioester bonds" better to put "via ester or thioester bonds"
line 56: "proteins than SP." correctly "proteins than IP."
line 117: "[53].Herein" a space is missing "[53]. Herein"
line 157-164: legend of Figure 2 is partly on next page, it would be better to keep it under the figure on the same page
Author Response
Reviewer #2
Comments and Suggestions for Authors
This is an excellent review, valuable summary on recent progress on the role of UPS in immune system. It has a logical structure, easy to follow explanations and comprehensive original figures. Chapter 5 about the contribution of UPS to certain diseases contains even a paragraph on the actual SARS-CoV-2.
A suggest a few minor corrections:
line 28: instead "ubiquitination is mediated by three groups of enzymes" write "ubiquitination is mediated by an enzyme cascade"
>> This has been changed.
line 38: "via oxyester or thioester bonds" better to put "via ester or thioester bonds"
>> This has been changed.
line 56: "proteins than SP." correctly "proteins than IP."
>> This has been changed.
line 117: "[53].Herein" a space is missing "[53]. Herein"
>> This has been changed.
line 157-164: legend of Figure 2 is partly on next page, it would be better to keep it under the figure on the same page
>> This has been changed.
Round 2
Reviewer 1 Report
All my concerns have been addressed in the modified version of this nice review.